# Analysis of intracellular and intercellular crosstalk from omics data

Alice Chiodi⬤, Paride Pelucchi⬤, Ettore Mosca⬤*

Institute of Biomedical Technologies, National Research Council, Segrate (Milan), Italy

* ettore.mosca@itb.cnr.it

## Abstract

Disease phenotypes can be described as the consequence of interactions among molecular processes that are altered beyond resilience. Here, we address the challenge of assessing the possible alteration of intra- and inter-cellular molecular interactions among processes or cells. We present an approach, designated as "Ulisse", which complements the existing methods in the domains of enrichment analysis, pathway crosstalk analysis and cell-cell communication analysis. It applies to gene lists that contain quantitative information about gene-related alterations, typically derived in the context of omics or multi-omics studies. Ulisse highlights the presence of alterations in those components that control the interactions between processes or cells. Considering the complexity of statistical assessment of network-based analyses, crosstalk quantification is supported by two distinct null models, which systematically sample alternative configurations of gene-related changes and gene-gene interactions. Further, the approach provides an additional way of identifying the genes associated with the phenotype. As a proof-of-concept, we applied Ulisse to study the alteration of pathway crosstalks and cell-cell communications in triple negative breast cancer samples, based on single-cell RNA sequencing. In conclusion, our work supports the usefulness of crosstalk analysis as an additional instrument in the "toolkit" of biomedical research for translating complex biological data into actionable insights.

## Introduction

The understanding of how gene-related molecular alterations translate into pathological phenotypes is a major challenge in life sciences. The experience gained from reductionist approaches like genome-wide association studies – where millions of single nucleotide variations are independently tested for association with a phenotype – strengthen the key role of molecular *interactions* for deciphering the roots of complex phenotypes [1]. The term "network medicine" was introduced to designate the application of network science to study diseases, which are viewed as the consequence of molecular alterations on a complex system of interacting molecular processes [2].

**Data availability statement:** The single-cell RNA sequencing data that support the findings of this study are available in "figshare" with the identifier https://doi.org/10.6084/m9.figshare.17058077.v1.

**Funding:** This work was supported by Fondazione Regionale per la Ricerca Biomedica (Regione Lombardia) (ERAPERMED2018-233 FindingMS GA 779282), European Commission H2020 GEMMA project (ID 825033), and European Union NextGenerationEU, Mission 4 Component 2, project "Strengthening BBMRI.it", CUP B53C22001820006. The funders had no role in study design, data collection and analysis, decision to publish, or preparation of the manuscript.

**Competing interests:** The authors have declared that no competing interests exist.

At cellular level we can classify molecular interactions into two broad categories: *intra*-cellular and *inter*-cellular, according to whether they take place *within* a cell or *among* cells. Even if our knowledge of intra- and inter-cellular molecular interactions is incomplete, molecular networks are a crucial tool in biomedical research to translate complex molecular data from omics and multi-omics studies into actionable results [3–5].

Here, we describe an approach, designated as "Ulisse", to gain mechanistic insight from gene lists that contain quantitative information about gene-level changes (e.g., mutated genes, differentially expressed genes). The method screens a collection of gene sets (e.g., protein complexes, molecular pathways, cell phenotype markers) to evaluate whether the interconnectivity between any gene set pair is more affected by gene-level changes than would be expected by chance. Considering the complexity of statistical assessment of network-based analyses, the statistical assessment is based on two complementary null models, which probe the outcome by sampling alternative configurations of gene-level changes and molecular interactions [6]. The main applications of Ulisse are in pathway analysis and cell-cell communication, where it can be used, respectively, to identify pathway crosstalk and cell-cell interactions that are affected by gene-level alterations detected through omics experiments (in bulk or at single-cell resolution). The analysis of pathway crosstalk and cell-cell communication can be combined to obtain integrated pathways that associate cell-cell communications and intracellular states. Lastly, the list of significantly altered crosstalks is analysed to identify the key genes that affect the interconnectivity among gene sets.

The applications of our approach encompass the domains of enrichment analysis, pathway crosstalk analysis and cell-cell communication analysis. Several approaches exist to perform such tasks, with relevant differences that stem from the specific focus for which methods are designed, input data types, outputs, statistical assessment of the results, and implementation.

Ulisse provides a novel type of analysis in comparison to existing gene set enrichment analysis approaches, which, net of some nomenclature variations, can be classified into four categories [7–9]: (i) over representation analysis or overlap methods (based on the hypergeometric test); (ii) functional class scoring or per-gene score analysis (e.g., GSEA [10] and GSVA [11]); (iii) pathway-topology based methods or pathway topology analysis (e.g., SPIA [12]); (iv) network crosstalk tools or network enrichment methods (e.g., ANUBIX [8] and NEAT [13]). Ulisse could be fit into the latter category, but it has a different focus. Indeed, existing network crosstalk tools evaluate the interconnectivity between a gene list (e.g., differentially expressed genes) and every gene set of a collection (e.g., KEGG pathways [14]), in the context of an interactome composed of gene-gene interactions (e.g., from STRING [15]). Instead, Ulisse quantifies to which extent the interconnectivity of every pair of gene sets (of the collection) is altered based on the information given in the input gene list.

The availability of computational tools to study pathway crosstalk is limited, despite their importance in regulatory mechanisms [16,17], obtaining effective drug combinations in cancer [18] and investigating complex diseases phenotype [19]. Two

examples are the Latent Pathway Identification Analysis (LPIA) [20] and Pathway analysis using Network information (PathNet) [21], both developed for gene expression data. LPIA defines a network of pathways based on shared Gene Ontology [22] terms and a list of differentially expressed genes. Then, it identifies the pathways with the most significant centrality in the network, using gene expression data permutations to create a null model. As such, the pathway crosstalk in LPIA does not consider the molecular interactions between the two pathways, which, instead, is what Ulisse focuses on. PathNet "contextual analysis" quantifies pathway crosstalk similarly to Ulisse. It requires gene p-values as input and provides a matrix of p-values that estimate the significance of pathway-pathway interconnectivity, where input gene p-values are shuffled to create a null model. Differently from PathNet, Ulisse focuses on the interconnectivity between the genes that are not shared by the two pathways, tests the significance of crosstalk against permutations of gene-level changes and gene-gene interactions, provides a more extensive output (e.g., crosstalk score, p-values, number of altered interactions, etc.), and identifies gene scores based on their relevance as mediators of pathway crosstalks.

As for the domain of cell-cell communication, recent advancements in sequencing technologies have prompted the development of quite a relevant number of tools and resources for cell–cell communication inference, denoted by wide variations in requirements, scoring approaches, type of communication inferred, assumptions, and limitations [23]. In this complex scenario, the peculiar features of Ulisse are the focus on reconstructing a communication network where cell states (or types) are nodes, the statistical assessment based on two different null models, the integration of pathway crosstalk analysis with cell-cell communication and the identification of key genes involved in the communication network.

In the following, we describe Ulisse and, as a proof-of-concept, we apply our approach to study the alteration of pathway crosstalks and cell-cell communications in publicly available single-cell RNA (scRNA) expression data from a recent study that proposed a high-resolution map of cell diversity in normal and cancerous human breast [24,25].

## Results

### Crosstalk quantification, statistical assessment and key players

Here we describe how we define an *altered* "crosstalk" (or *active* crosstalk) between two gene sets, the assessment of its statistical significance and, lastly, how we use the results of crosstalk analysis to score genes based on their contribution (Fig 1, see S1–S3 Text in S1 File for further details). Note that we adopt a "gene-centric" view of molecular interactions – like in gene-centric human interactomes [4,5] – where the term "gene-gene interaction" refers to various types of molecular interactions (protein-protein, protein-RNA, protein-DNA) that involve the considered gene pair.

Two types of input are needed to calculate the altered crosstalk between any two gene sets X and Y (Fig 1a):

- a list of gene-gene interactions, which can be derived from publicly available resources (like STRING [15] and Omnipath [3,26]);

- one or more sets of gene-level weights (in the unit interval), which provide a summary of the gene-level alterations of interest (e.g., differential expression or mutations).

Formally, we quantify the crosstalk score as the sum of weighted products between the genes of X that interact with those of Y:

$$C(X, Y) = \mathbf{u}_X^\mathsf{T} \mathbf{A} \, \mathbf{u}_Y = \sum_{i=1}^{N_G} \sum_{j=1}^{N_G} a_{ij} \, \mathbf{u}_X(i) \mathbf{u}_Y(j),$$

where $\mathbf{A} = (a_{ij})$ is the adjacency matrix that specifies the interactions among the $N_G$ genes, while $\mathbf{u}_X$ and $\mathbf{u}_Y$ are vectors of gene weights with positive values only for the genes of X, Y.

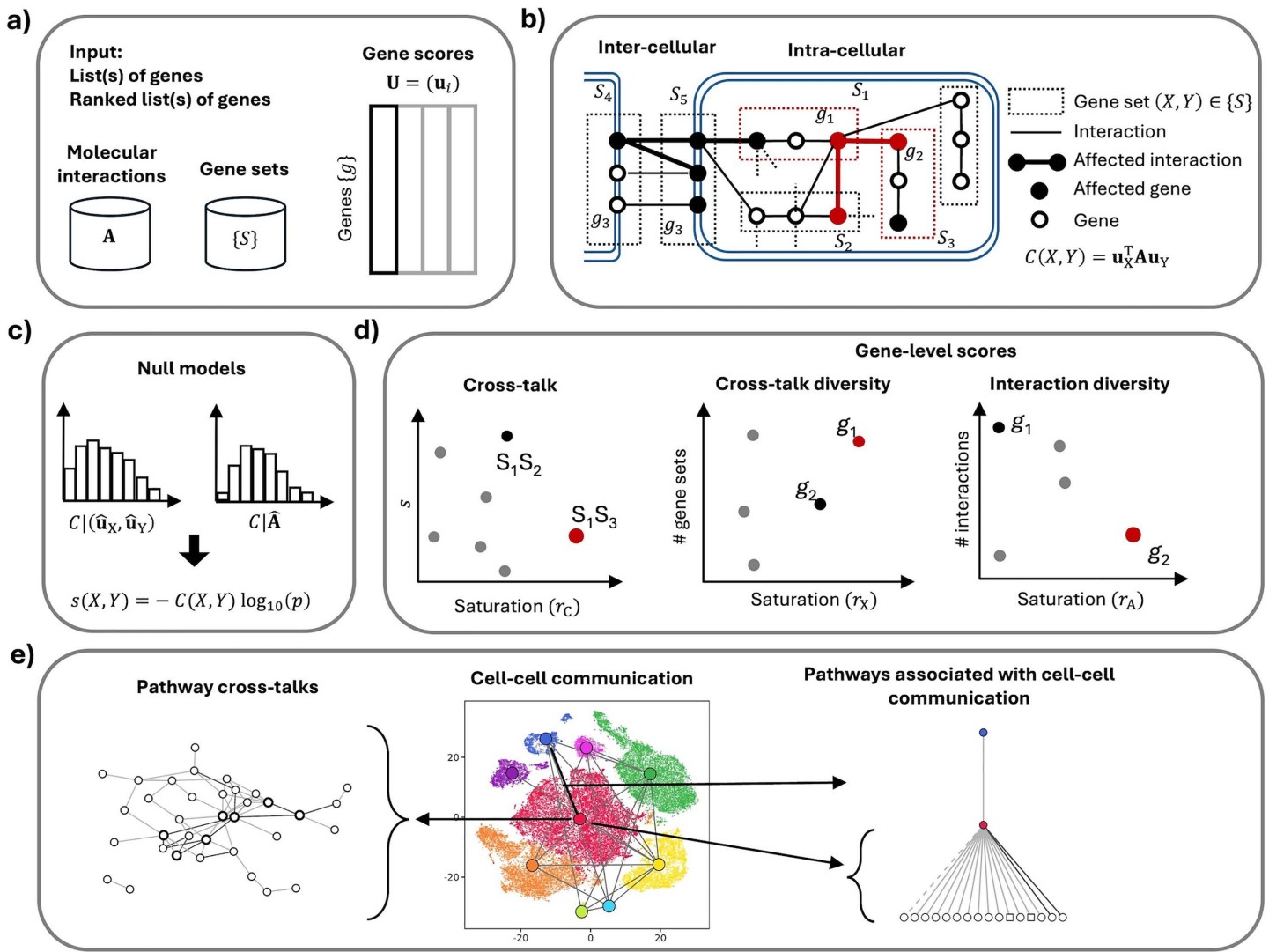

**Fig 1. Overview of crosstalk analysis. a)** Input data. **b)** Visualization of the molecular interactions among gene sets, at inter-cellular and intra-cellular levels, which lead to altered crosstalks. **c)** The crosstalk value is supported by two null models. **d)** Crosstalk values can be distinguished based on their saturation; the crosstalk diversity and interaction diversity are two gene-level scores that enable the identification of key crosstalk mediators; these scores can be distinguished based on their saturation. **e)** Crosstalk analysis identifies networks of intra-cellular processes, cell-cell communication and intra-cellular processes associated with cell-cell communication.

The definition of the quantities involved in the calculation $C(X, Y)$ follows three scenarios, based on whether the crosstalk is between gene sets associated with intra-cellular states, inter-cellular states or both (Fig 1b).

To quantify the alteration of crosstalk between two *intra-cellular* pathways (or another type of gene system), we exclude the genes shared between them, that is $X \cap Y = \varnothing$, otherwise we would consider intra-pathway interactions. Further, gene weights $\mathbf{u}_X$ and $\mathbf{u}_Y$ are defined from the same input source (e.g., same set of alterations), because the focus is on an intra-cellular characteristic, and the two gene sets represent internal states of the same cell. The molecular interactions are collected from "general purpose" database of interactions, like STRING [15].

 

Conversely, to quantify the active *inter-cellular* crosstalk between two gene sets that are associated with two cell types, it is expected to have $X \cap Y = \varnothing$, because the two cell types, for example, can express a series of genes in common. Moreover, $\mathbf{u}_X$ and $\mathbf{u}_Y$ come from different sources, because the alterations are relative to distinct cell types. The molecular interactions are collected from databases that focus on ligand-receptor, like Omnipath [3,26], a collection composed by multiple sources (i.e., Ramilowski [27], CellPhoneBD [28]).

Lastly (third scenario), to quantify the intra-cellular alterations associated with inter-cellular alterations, we consider – for each cell type under analysis – the genes (set $X$) involved in any of the inter-cellular crosstalks of the cell type, and any altered pathway (set $Y$) of the cell type. Besides such peculiarity in the definition of $X$ and $Y$, we have, like in the first scenario, that $X \cap Y = \varnothing$, and $\mathbf{u}_X$ and $\mathbf{u}_Y$ are defined from the same input source, because they are relative to the same cell type.

A meaningful quantity that complements $C(X, Y)$ is the crosstalk *saturation*

$$r_C = \frac{\delta L_{XY}}{L_{XY}},$$

which captures, in the process under study, the number of altered interactions $\delta L_{XY}$ between $X$ and $Y$, in relation to all the possible interactions $L_{XY}$ between $X$ and $Y$. Indeed, similar values of $C(X, Y)$ can be due to a higher or lower impairment of the links between the two gene sets.

To statistically benchmark the magnitude of an observed crosstalk value $c$ we have to consider that it might depend on various features like gene set size, distribution of gene weights and gene degree. We focused on two null models, namely $M_A$ and $M_u$, in both of which we preserve gene set size, degree sequence, and the association between gene weight and gene degree (within the same bin over the degree sequence), and randomize, respectively, gene-gene interactions and gene weights ([Fig 1c]). Null model $M_A$ is designed to test the dependence of $c$ from the network proximity of $X$ and $Y$, while $M_u$ is meant to test the dependence of $c$ from the weights of $X$ and $Y$ genes. This leads to four possible outcomes, determined by the possible significance of $c$ in a single null, in both or in neither of them (S1 Fig in [S1 File], S2 Table in [S2 File]). As expected by the fact that the two nulls disrupt different features, the analyses performed in our proof-of-concept (see next sections) revealed negligible correlations between the values obtained with the two nulls (S2 Fig in [S1 File]). It is therefore meaningful to combine the two probabilities $\rho_A$ and $\rho_u$, of observing – respectively – a value equal or greater than $c$ in $M_A$ and $M_u$, into the probability of observing a product $\hat{\rho}$ as small as the one observed

$$p = P\left(\hat{\rho} \leq \rho_A \rho_u\right) = \rho_A \rho_u - \rho_A \rho_u \ln\left(\rho_A \rho_u\right),$$

which is equal to the probability obtained by means of the so-called Fisher's combined probability test [29,30].

Lastly, we define a summary score for ranking crosstalks, combining effect size $C(X, Y)$ and its estimated probability $p$:

$$s(X, Y) = -C(X, Y) \log_{10}(p)$$

The list of significant crosstalks provides an opportunity for gene scoring ([Fig 1d]). We consider two gene-level quantities: *crosstalk diversity* and *interaction diversity*. The first counts how many gene sets that are part of altered crosstalks contain interactors of a gene $g_i$; therefore, the saturation $r_X(i)$ of the crosstalk diversity of $g_i$ reaches 1 when all the gene sets that contain interactors of $g_i$ are part of altered crosstalks. The second counts the interactors of $g_i$ that belong to gene sets that are part of crosstalks; analogously to $r_X(i)$, the saturation $r_A(i)$ of the interactor diversity of $g_i$ reaches 1 when all the interactors of $g_i$ belong to gene sets that are part of altered crosstalks.

## Alteration of crosstalks in triple negative breast cancer

As a proof-of-concept, we analysed the crosstalks in triple negative breast cancer (TNBC), using single-cell RNA expression data from a recent study that proposed a high-resolution map of cell diversity in normal and cancerous human breast

[24,25]. Our objective is to show what kind of information can be extracted from the analysis of crosstalks using data generated by means of one the state-of-the-art technologies in transcriptomic analysis. In particular, we focused on cancer cells and analysed the interactions among intra-cellular processes whose alteration could be implied in the dysregulation of the reciprocal control among molecular mechanisms that could contribute to tumour progression. Then, we considered the communication between cancer cells and Cancer Associated Fibroblasts (CAF). Indeed, CAFs represent a peculiar hub of cell-cell communication within the tumour niche by promoting tumoral growth and malignancy by releasing factors targeting cancer cells, repressing immune response by their interactions with immune cells and inducing angiogenesis interacting with endothelial cells [31–33]. Lastly, we shed light to cancer cell processes that could be associated with the communication between CAFs and cancer cells.

**Alteration of intra-cellular crosstalks in cancer cells.** We screened the role of 304 (S2 and S3 Tables in S2 File) cancer epithelial cell markers (FDR < 0.05, $\log_2$(FC) > 0.5, Cancer epithelial vs all) in the crosstalk among intra-cellular processes (MSigDB Hallmarks database [34]). We found that most of the crosstalks is altered in up to 2 interactions and that the maximum number of altered interactions is 19, among a total of 14 genes belonging to allograft rejection and *MYC* targets (v1) (Table 1, Fig 2a, S4 Table in S2 File). We observed a marked variability of gene weights, degree of statistical significance, and saturation, independently from the number of affected links, which makes such pieces of information useful to differentiate crosstalks (Fig 2a–2c). In particular, we observed a total of 59 crosstalks whose score can hardly be obtained ($\alpha = 0.01$) when shuffling interactions or gene weights, and 14 crosstalks that are supported by both nulls (Fig 2b). Among these, we obtained several crosstalks that involves the p53 pathway, cholesterol homeostasis and androgen response, which emerge as hubs in the network of altered crosstalks (Fig 2d). In 24 crosstalks, the saturation indicates the alteration of more than half of the links (Fig 2c), like between p53 pathway and KRAS signalling ("KRAS_SIGNALING_DN"), which involves the alterations of 3 out of 4 interactions between a total of 5 genes, and the score is supported by both nulls (Fig 2c, S4 Table in S2 File).

To compare the outcomes of crosstalk and pathway enrichment analyses, we assessed to which extent the processes exhibiting significant crosstalks are also marked by significant enrichment (hypergeometric test) in DEGs (Fig 2e, S5 Table in S2 File). As expected, the two types of analyses provide a complementary view, where several processes involved

**Table 1. Top 5 pathway crosstalks mediated by cancer cell DEGs.**

| X | Y | $|X|$ | $|Y|$ | $\|u_X\|$ | $\|u_Y\|$ | $\delta L_{XY}$ | $L_{XY}$ | $c$ | $r_c$ | $\rho_A$ | $\rho_u$ | $p$ | $s$ |
|---|---|---|---|---|---|---|---|---|---|---|---|---|---|
| ALLOGRAFT REJECTION (*RPS19, CDKN2A, RPL39*) | MYC TARGETS V1 (*FBL, MYC, PABPC1, RPL18, RPL6, RPLP0, RPS10, RPS2, RPS3, RPS5, RPS6*) | 62 | 53 | 0.338 | 1.334 | 19 | 101 | 0.252 | 0.188 | 0.001 | 0.001 | 1.48E-05 | 1.219 |
| ESTROGEN RESPONSE EARLY (*CCND1, MYC, KRT15*) | P53 PATHWAY (*CDKN2A, CDKN2B, KRT17*) | 44 | 50 | 0.593 | 0.404 | 5 | 19 | 0.120 | 0.263 | 0.001 | 0.001 | 1.48E-05 | 0.578 |
| KRAS SIGNALING DN (*KRT15, KRT5, PKP1*) | P53 PATHWAY (*KRT17, SERPINB5*) | 14 | 53 | 0.584 | 0.304 | 3 | 4 | 0.110 | 0.750 | 0.001 | 0.001 | 1.48E-05 | 0.532 |
| ANDROGEN RESPONSE (*DBI, KRT19, KRT8*) | APOPTOSIS (*APP, CLU, KRT18*) | 29 | 56 | 0.665 | 0.395 | 4 | 7 | 0.109 | 0.571 | 0.001 | 0.001 | 1.48E-05 | 0.528 |
| MYC TARGETS V1 (*MYC, FBL, RPL6, RPLP0, RPS10, RPS2, RPS3, RPS5, RPS6*) | P53 PATHWAY (*CDKN2A, CDKN2B, RPS12*) | 53 | 52 | 1.155 | 0.292 | 10 | 27 | 0.116 | 0.370 | 0.002 | 0.001 | 2.82E-05 | 0.526 |

The genes reported between parentheses are the DEGs that contribute to the crosstalk. The notation |.| indicates gene set size, while ‖.‖ indicates the sum over all gene weights that contribute to the crosstalk.

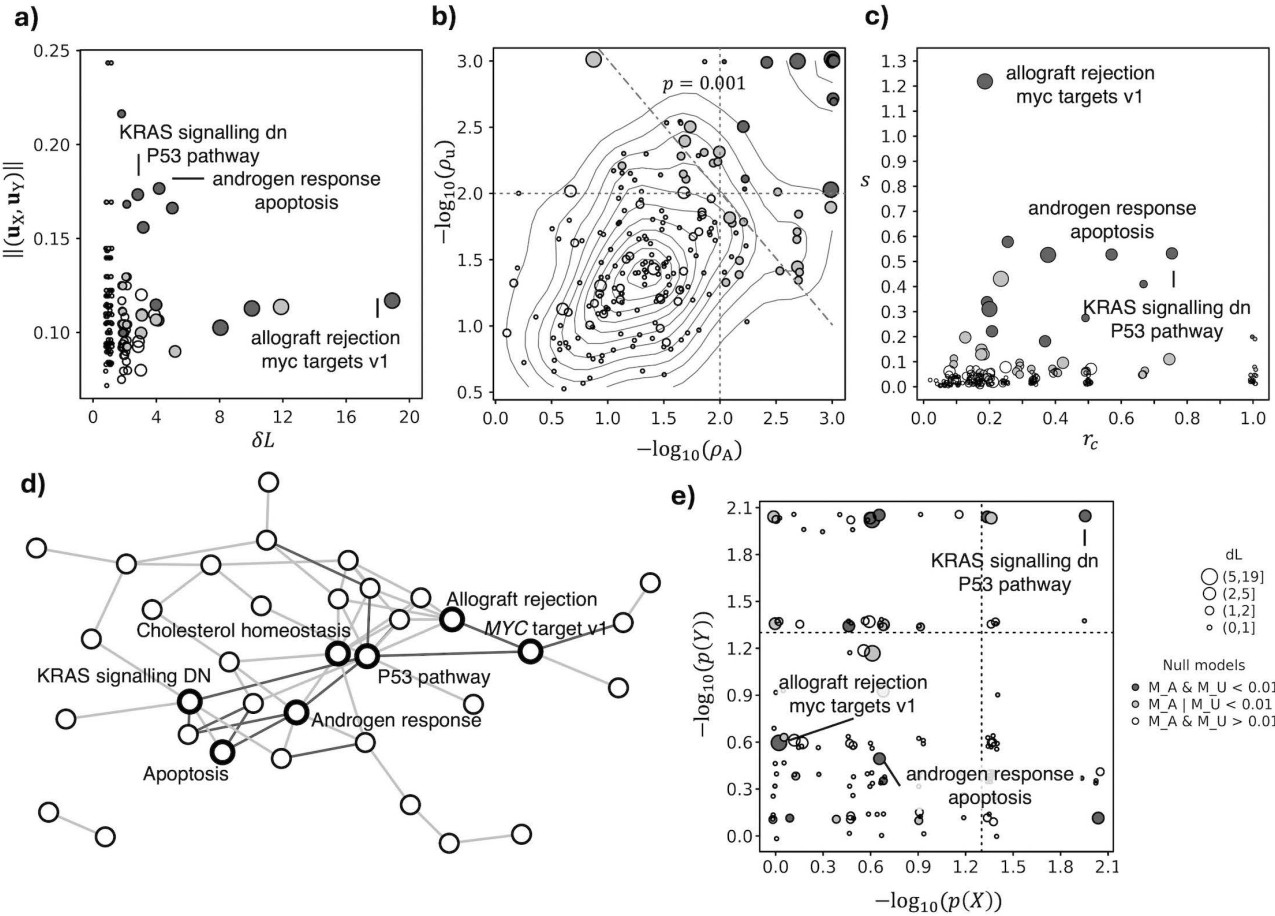

**Fig 2. Intra-cellular crosstalks controlled by expression changes specific of TNBC cells.** **a)** number of altered links ($\delta L$) and average gene weight $\|(\mathbf{u}_X, \mathbf{u}_Y)\|$ of the crosstalk forming genes. **b)** The probabilities $\rho_A$ and $\rho_u$ estimated by the two null models for each crosstalk value; the vertical and horizontal lines denote $\alpha = 0.01$, while the diagonal line denotes $p = 0.001$. **c)** Crosstalk score $s$ and its saturation $r_c$. **d)** Network of processes that establish crosstalks supported ($\alpha = 0.01$) by at least a null model. **e)** Over representation analysis p-values $p(X)$ and $p(Y)$ for each of the processes ($X$, $Y$) that establish a crosstalk; the vertical and horizontal lines denote $\alpha = 0.05$.

in significant crosstalks do not display enrichment and *vice versa*. Only in a few cases (7 pairs) both the processes are enriched (*p*-value < 0.05) in DEGs, while the majority of altered crosstalks takes place between pairs of processes that are not enriched in DEGs.

Significantly altered crosstalks are mediated by a total of 57 genes (S6 Table in S2 File). Crosstalk diversity and interaction diversity suggest a gene prioritization that is independent from their initial alteration score. In other words, genes that were ranked low by differential expression analysis can emerge as key players as mediators of crosstalks. This is the case of the two cyclin-dependent kinases *CDKN2A* and *CDKN2B,* which stand out for their crosstalk diversity, as they mediate 11 and 10 altered crosstalks, respectively (Fig 3a). Among the genes with the highest interactor diversity, we obtained a series of genes that code for ribosomal-associated proteins (Fig 3b). We observed a wide range of saturations and an overall correlation between the saturation of crosstalk diversity and that of interaction diversity (S6 Table in S2 File). Among the genes with the highest values of both saturations we found the two tumour proteins D52 (*TPD52*) and D53 (*TPD52L1*) (Fig 3), which are involved in cancer cells proliferation and more aggressive phenotype [35,36].

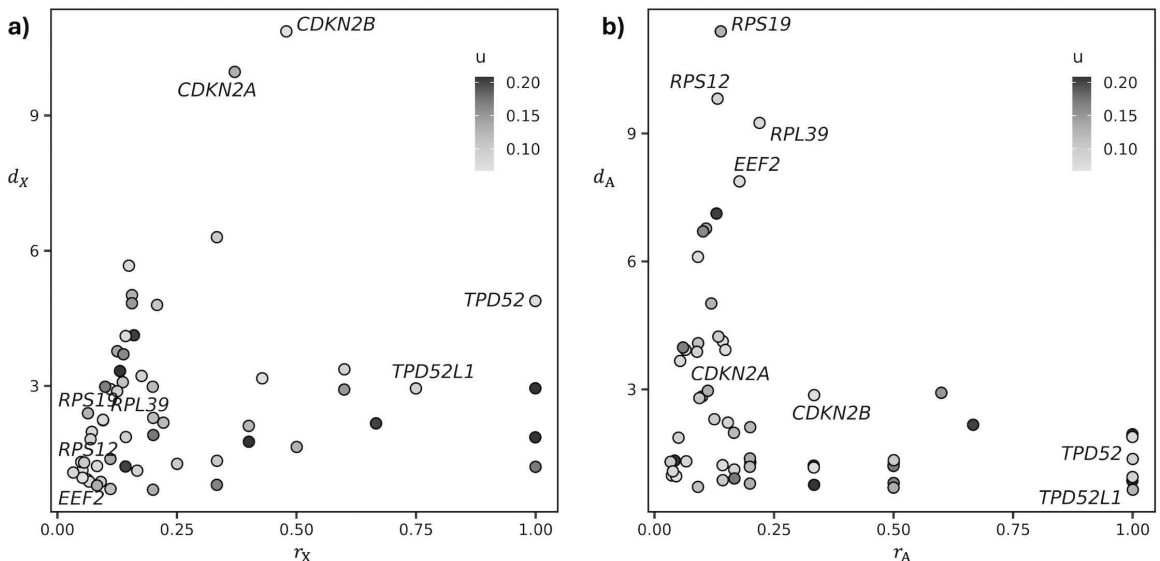

**Fig 3. Crosstalk diversity and interaction diversity of the DEGs that are involved in intra-cellular crosstalks in TNBC cells. a)** Crosstalk diversity $d_X$ and its saturation $r_X$. **b)** Interactor diversity $d_A$ and its saturation $r_A$.

**Inter-cellular crosstalks (cell-cell communication).** We analysed the inter-cellular interactions (Omnipath [3]) among the 36 pairs of gene sets defined by the DEGs (FDR < 0.05, $\log_2$(FC) > 0.5) of 9 cell types (1-vs-all), in 8 TNBC tumors [24] (S2 and S3 Tables in S2 File). Compared to the intracellular crosstalks among processes, here we dealt with larger gene sets and more links among them. As expected, this scenario led to a higher number of altered interactions, with a median of 38 and a maximum of 162 between CAFs and endothelial cells (Fig 4a, S7 Table in S2 File). Further, the crosstalks are statistically supported ($\alpha$ = 0.01) mostly by their gene weights (21 pairs), rather than interactions, which support the 3 crosstalks that are significant in both nulls, namely between B cells and tumour-associated macrophage (TAMs), between dendritic cells (DCs) and TAMs, and between B cells and DCs. Saturation reaches up to one quarter of the possible interactions between CAFs and endothelial cells (Fig 4b).

The emerging cell-cell communication network (Fig 4c) highlights a relevant role of such microenvironment cells, which establish several significant interactions. The communication between cancer cells and CAFs is supported ($\alpha$ = 0.01) by randomization of gene weights, and involves 22 interactions between a total of 33 DEGs (Fig 4c, S8 Table in S2 File). Among the key players of this communication, we found *MDK* and *MFGE8* (expressed in cancer cells), which mediate 4 and 3 interactions, respectively, with genes expressed in CAFs, including integrins *ITGB1* and *ITGB5* (S8 Table in S2 File).

The cell-cell communication network ($\alpha$ = 0.01) involves 379 genes (Fig 5, S9 Table in S2 File). Among the genes that stand out for their ubiquity we observed *CXCR4*, with a crosstalk diversity of 8 (out of 9 cell-types present), and *ICAM1*, *TGFB1*, *ITGB2*, *PTPN6* and some Major Histocompatibility Complex genes (*HLA-C*, *HLA-DRA*, *HLA-DRB1*), which show a crosstalk diversity of 7. Among the 29 DEGs in cancer epithelial cells (out of 379), *MFGE8*, *LAMP1*, *RPSA* and *AZGP1* are specific ($d_X$ = 1, $r_X$ = 1) of the communication with CAFs (Fig 5). Conversely, we did not observe DEGs in CAFs that are specific to the signalling with cancer cells. However, there is one gene, *PLAT*, which is uniquely involved in the communication with cancer cells ($d_X$ = 1).

**Integrated crosstalks: Cancer cell pathways that can be associated with the communication between cancer cell and CAFs.** To identify cancer cell pathways that can be associated with the communication between cancer cell and CAFs, we analysed the crosstalks between the gene set of the 14 cancer cell DEGs that mediate

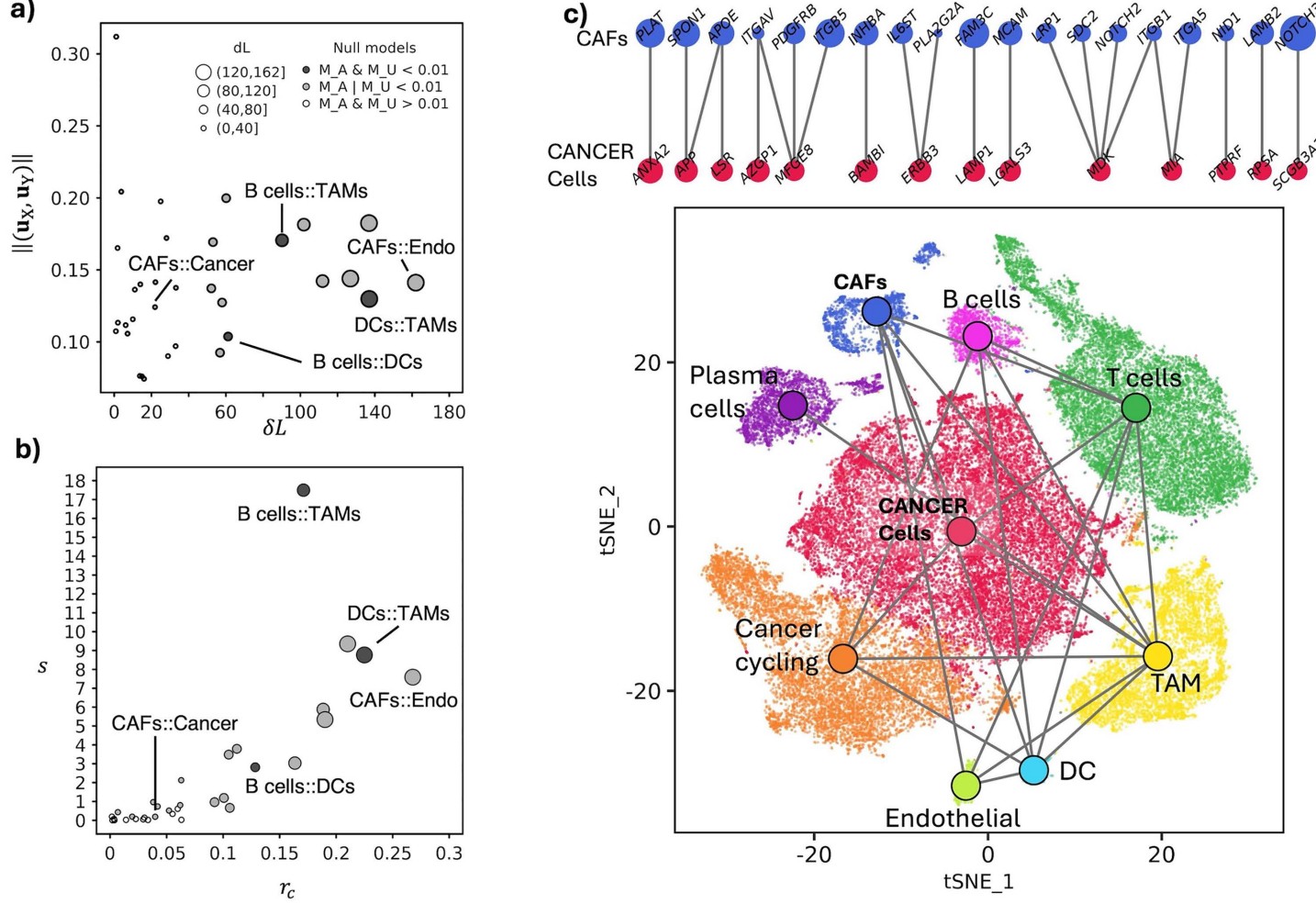

**Fig 4. Inter-cellular crosstalks controlled by expression changes in the 9 cell types of TNBC samples. a)** Number of altered links ($\delta L$) and average gene weight ($\|(\mathbf{u}_X, \mathbf{u}_Y)\|$) of the crosstalk forming genes. **b)** Crosstalk score $s$ and its saturation $r_c$. **c)** Above: DEGs that mediate the communication between CAFs and cancer cells; below: the position of cells in the space of the first two tSNE dimensions (bottom), coloured by cell type whose communications supported ($\alpha = 0.01$) by at least a null model are indicated through a link between the two centroids.

the communication with CAFs (CAFs-Cancer cell communication), and cancer cell pathways (MSigDB Hallmarks) that contain cancer cell DEGs (Fig 6, S10 Table in S2 File). We found 15 interactions supported ($\alpha = 0.01$) by at least a null model and two supported by both nulls. The first involves interactions among *RPSA* (CAFs-Cancer cell communication), and other ribosomal proteins (*RPL18*, *RPL6*, *RPLP0*, *RPS10*, *RPS2*, *RPS3*, *RPS5*, *RPS6*) that are regulated by *MYC*. The second take place between, *APP* and *PTPRF* (CAFs-Cancer cell communication), and *CLU* and *CTNNB1* (cholesterol homeostasis).

Almost all the interactions found (13 out of 15) involve processes that establish significant ($\alpha = 0.01$) intra-cellular crosstalks in cancer cells (S4 Table in S2 File). Conversely, the pathways of complement and coagulation did not emerge in the screening of intra-cellular crosstalks in cancer cells. Their relation with CAFs is mediated by the interaction between *APP* (CAFs-Cancer cell communication) and *CLU* (complement and coagulation).

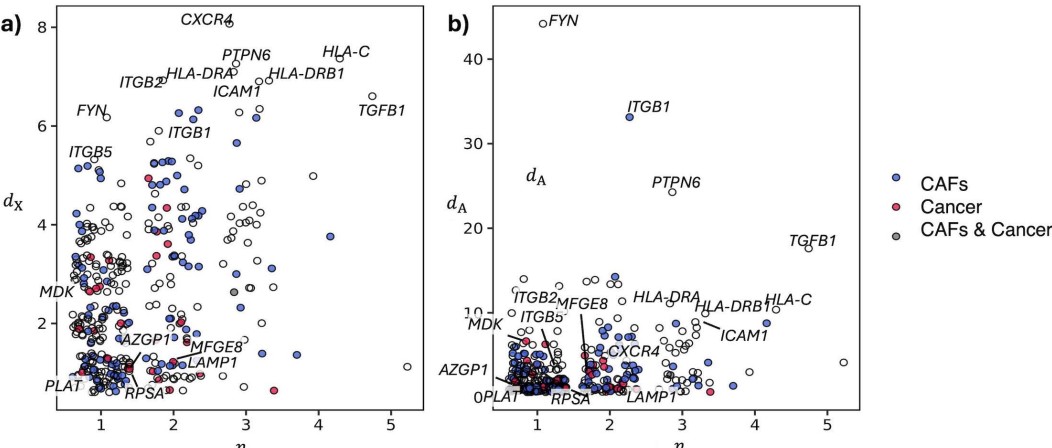

**Fig 5. Crosstalk diversity and interaction diversity of the DEGs that are involved in inter-cellular crosstalks between CAFs and cancer cells.**
**a-b)** Crosstalk diversity $d_X$ **(a)** and Interactor diversity $d_A$ **(b)** in relation to the number of cell types ($n$) in which the gene is differentially expressed.

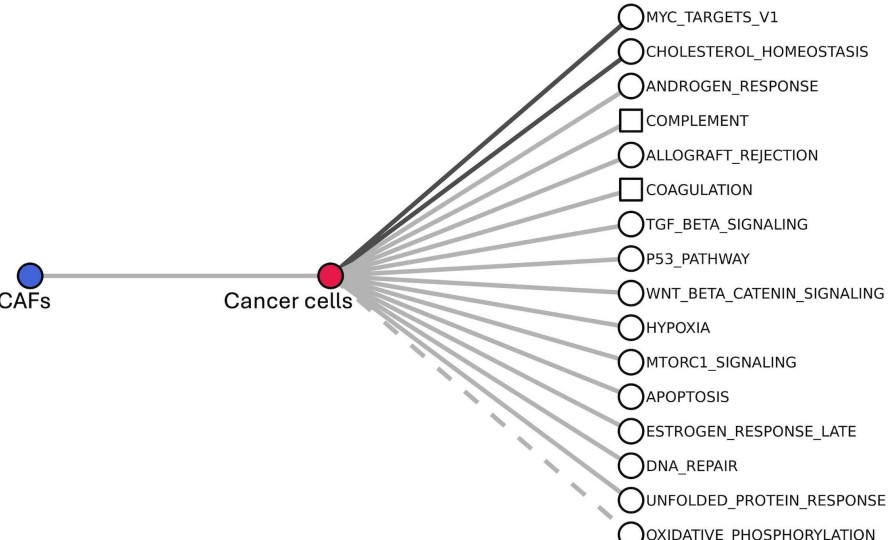

**Fig 6. Intra-cellular processes of cancer cells associated with their signalling with CAFs.** The processes are ranked from top to bottom by decreasing value of $s$; squares indicate processes that were not found in the analysis of intra-cellular crosstalks; link colour indicates statistical evidence (as in Figs 2 and 4), with the exception that, here, the dashed line replaces the white colour in indicating that both null models are above $a = 0.01$.

## Discussion

We presented a network-based approach (Ulisse) to assess the alterations of crosstalks between gene sets, based on gene-centric molecular interactions and one or more lists of gene scores that result from omics data analysis. According to how gene sets, gene lists and gene-gene interactions are defined, our method can be applied to inter-cellular as well as intra-cellular crosstalks, and to the analysis of intra-cellular crosstalks that can be associated with inter-cellular crosstalks. The score of a crosstalk is proportional to the gene-gene interactions between the two gene sets that are affected by the gene-level changes. This provides an intuitive means to quantify crosstalks, which can be understood in terms of the

interactions and molecular alterations involved. As a proof-of-concept, we analysed the crosstalks affected by the gene expression alterations detected at single-cell resolution in triple negative breast cancer samples, because this type of data allowed us to demonstrate the two main applications, that is, pathway crosstalk analysis and cell-cell communication analysis. However, the method is not restricted to gene expression data. The pathway crosstalk analysis can be applied to gain insights from various types of gene lists (e.g., containing information on DNA variation), derived from new experiments as well as from public resources, such as, in the domain of cancer, the Genomic Data Commons [37], COSMIC [38] or the cBio Cancer Genomics Portal [39]. Similarly, the cell-cell communication analysis in Ulisse can also handle other data types, like mutational or epigenetic profiles at single cell resolution.

The score is supported by two complementary null models that conserve gene set size, degree sequence, and the association between gene weight and gene degree. These nulls provide a means to assess whether the statistical significance of the score comes from gene weights, interactions or both. In the proof-of-concept, we showed that all three scenarios emerge when using real data, underlying the importance of assessing the crosstalk analysis outcome from different angles [6].

We reported altered crosstalks at various degree of saturation, especially in the analysis of intra-cellular crosstalks. This quantity enabled the identification of pairs of processes where most of the interactions involved gene expression changes or, on the opposite, pairs of processes where only a specific part of their interaction is affected. For example, our analysis identified that the pathway of KRAS is involved in crosstalk dysregulation associated with TNBC, supporting evidence that indicates this pathway as crucial in phenotypical and metabolic features of cancer cells [40].

We showed that the analysis of intra-cellular crosstalks complements the typical enrichment analysis. Indeed, we reported a series of gene sets that, despite not showing significant enrichments in DEGs, were part of significantly altered crosstalks. This is the case of one of the top ranked crosstalks (supported by both nulls), which suggests the impairment of regulative mechanisms between androgen response and apoptosis. Notably, the relation between androgen receptor and apoptosis has been implicated in breast cancer metastasis [41,42]. Another example is cholesterol homeostasis, which emerged as a hub of the intra-cellular network and is reported to promote cancer cell proliferation in TNBCs [43].

The reconstruction of inter-cellular communications based on cell type-associated gene sets provides a means to overcome the heterogeneity at gene expression level and sheds light on the general picture of the active (or altered) communications among the cell types. The analysis of TNBC cell types confirmed the well-known core network of communications between cancer cells and microenvironment. Cancer cells show significant communication with CAFs, supporting the pro-tumoral role of CAFs by activating the signalling associated with proliferation and tumour progression [31–33,44].

The joint analysis of intra-cellular and inter-cellular cross talks paves the way towards the reconstruction of maps that integrate the communications between different cell types with the pathway crosstalks activated within each one. In the proof-of-concept we analysed the processes that are activated in cancer cells and can be associated with their communication with CAFs. Notably, a mediator of such crosstalk is the extracellular chaperone *CLU*, which was reported as a key player in cancer [45] and an interesting actionable target in TNBC [46,47].

With the aim of providing an additional way of identifying the genes associated with a phenotype, we introduced the crosstalk diversity and interaction diversity. These quantities, which are fundamentally different from purely topological features like degree and betweenness centrality, shed light on the genes that act as mediators of the signalling between processes or cell types. We showed, as a proof-of concept, that several genes with extreme crosstalk diversity and interaction diversity are indeed known to be associated with the process under study. Among them, *EEF2* was demonstrated to be upregulated in several cancers and associated with worse prognosis, thus suggesting its potentiality as novel therapeutic target [48,49]. The high interaction diversity of ribosomal-associated proteins sustains the importance of the dysregulation of translation process in tumorigenesis mechanisms and the clinical potential represented by targeting this process in tumour cells [50]. Interestingly, some of the genes prioritized by crosstalk diversity and interactor diversity have marginal expression changes and, therefore, stand out due to their pattern of interactions with other altered genes. This is the case of *CDKN2A* and *CDKN2B,* which exert a role in the regulation of cell cycle and proliferation and their association with breast cancer is largely studied [51,52]. The analysis of genes that mediate

the inter-cellular communications revealed a series of genes shared by multiple communications. These genes are involved in tumour promoting functions supporting tumour growth, chronic inflammation and angiogenesis, by secretion of growth factors and other soluble molecules, vesicles, and mechanic interactions among cells and extracellular matrix [33,53–55]. Concerning the genes that mediate the signalling between CAFs and cancer cells, *MDK* and *MFGE8* (expressed in cancer cells) are known to be associated with the acquisition of various tumour hallmarks [56,57]. Studies suggest the involvement of *AZGP1* in the differentiation of progenitor cells into CAF to support tumorigenesis [58], while *RPSA* and *LAMP1* are implicated in poor prognosis in breast cancer [59,60]. *PLAT* was reported to regulate the ability of breast cancer CAFs to invade stroma [61], and as an angiogenetic factor of CAF associated with negative prognosis in colon cancer [62].

Crosstalk diversity and interaction diversity can be relevant for the choice of actionable targets. Genes that affect several crosstalks interacting with multiple cellular functions are interesting targets for therapy, but – at the same time – could be associated with a wide spectrum of negative side effects. The saturations of crosstalk diversity and interaction diversity provide a means to collect more selective targets for therapy, as it prioritizes genes that mediate crosstalks with less but more disease-specific cellular functions.

The results presented in this study have to be seen in light of some limitations. The gene-gene interactions available in the literature are aspecific, and as such, they are a model of the interactions that *potentially* take place in the biological system under analysis. Moreover, the collections of molecular interactions are known to be affected by the various biases [4,5]. We have used state-of-the art collections and filtered the interactions to ensure an appropriate trade-off between coverage of genes and presence of biases, following the recommendations of previous studies [4,5]. As a proof-of-concept, we studied the intra-cellular crosstalks using gene set definitions from MSigDB hallmarks [34]. There are multiple ways to define intra-cellular processes, e.g., using databases like KEGG [14] and Reactome [63]. Therefore, other analyses of intra-cellular crosstalks in cancer cells of TNBC are possible and could highlight additional mechanisms. Moreover, the number of tested genes was limited by the sensitivity and depth of the scRNA sequencing technology underlying the data we used. In turns, the results emerged in the proof-of-principle should be interpreted considering this limited observability of the underlying molecular processes. Possibly, if the information on the input data is sparse, one could consider an amplification of the input gene weights based on the network proximity of genes, using a diffusion-based transformation [64,65].

Overall, Ulisse provides novel opportunities in the domains of gene set enrichment analysis, pathway crosstalk analysis and cell-cell communication. Collectively, compared to existing methods (see the introduction), Ulisse has a different focus, considers a more extensive statistical assessment, supports various input types and provides a different outcome. Such differences would lead to results that are more or less different from what can be obtained using other tools on the same input data. Therefore, considering such differences between Ulisse and existing tools and the issues in performing benchmarks in the absence of ground truths [23], we left a quantitative comparison for future work. Indeed, a trustworthy comparison among tools in the absence of a ground truth should consider multiple approaches, like literature agreement, experimental validation, indirect validation, synthetic dataset validation and robustness assessment truths [23]. Instead, in this manuscript, we focused on describing Ulisse, its relation to existing approaches, and showing the relevance of the results achieved in the proof-of-concept, considering the literature on breast cancer.

In conclusion, the approach presented in this work and the results gained in the proof-of-principle, even in the light of their limitations, support the usefulness of crosstalk analysis as an additional instrument to the "toolkit" of biomedical research for translating complex biological data into actionable insights.

## Methods

### Definition of gene weights from single cell RNA-sequencing data

The Seurat data object "SeuratObject_TNBC.rds" containing single-cell RNA expression data of 8 triple negative breast tumors [24,25] was downloaded from figshare [66]. The associations between the 9 cell clusters and cell types (not available in the Seurat object) were obtained on the basis of the cell association provided by the authors in the figures of the

paper, together with "SeuratObject_TNBCSub.rds" object (S3 Fig in S1 File). Differentially expressed genes were obtained by means of MAST algorithm [67], testing each cell type against all the other cells (Seurat [68] function "FindAllMarkers()", default parameters). Differential expression statistics were used to define a gene weight vector $\mathbf{u}_j$ (of size equal to the total number of genes in the considered analysis) for each cell type $j$ combining the log fold change $x_{ij}$ of each gene $i$ with its adjusted $p$-value (Benjamini-Hochberg method [69]); to reduce noise, scores associated with marginal significance were set to zero, that is $y_{ij} = -\log_2(x_{ij})\log_{10}(p_{ij})$ when $p_{ij} < 0.05$ and $\log_2(x_{ij}) \geq 0.5$, while $y_{ij} = 0$ otherwise. Each vector was normalized to have a maximum value of 1: $u_{ij} = y_{ij}/\max_i(y_{ij})$.

## Molecular interactions and gene sets

Molecular interactions used for pathway crosstalk analysis were downloaded from STRING [15] (v12, https://string-db.org/cgi/download). The combined score was updated excluding "text mining" using a modified version of the script "combine_subscores.v2.py" (https://stringdb-downloads.org/download). Ensembl identifiers were mapped to Entrez Gene identifiers using the mapping available in STRING (https://string-db.org/cgi/download) and Entrez Gene (ftp://ftp.ncbi.nih.gov/gene/DATA, September, 19, 2023). The highest score was considered for each gene pair. Only high-confidence (combined score ≥ 700) interactions and the top 3 (per gene) interactions with medium confidence (STRING score ≥ 400) were considered, obtaining a total of 174'962 interactions involving 17'288 genes. Molecular interactions available in Omnipath [26] were obtained through the R package OmnipathR [70] (September, 2024), for a total of 4'312 interactions involving 1'782 genes. The MSigDB Hallmarks gene sets [34] were collected through the R package "msigdbr" v7.4.1 [71].

In each analysis, the initial gene set list was created to ensure that: each gene had at least an interaction; only gene sets with at least 3 elements and a non-null gene weight were considered; to reduce the number of possible gene set pairs, only those such that $C(X, Y) > 0$ were considered.

## Randomizations and computational aspects

A total of 1000 randomizations of gene labels was used to create the null models. Gene degree was preserved splitting the degree sequence in equally sized bins, 9 for intra-cellular crosstalks, 4 to study inter-cellular communications, and 7 to study cancer cell intracellular crosstalks associated with their communication with CAFs. The number of bins was defined as the highest value (at most 15) that guaranteed non-empty bins. The average computational cost of the analysis of intra-cellular crosstalks (203 gene set pairs) was approximately 4 minutes on 8 cores with 64GB of RAM per core.

## Code availability

The computational method used in this study (Ulisse v2.0) is available in Zenodo with the identifier 10.5281/zenodo.15166722. Source code and documentation are freely available in github at the URLs https://github.com/emosca-cnr/Ulisse and https://emosca-cnr.github.io/Ulisse.

## Supporting information

**S1 File. S1-S3 Texts, S1-S3 Figures, S1 Table, and captions of S2-S10 Tables.**
(PDF)

**S2 File. S2-S10 Tables.**
(XLSX)

## Author contributions

**Conceptualization:** Ettore Mosca.

**Data curation:** Alice Chiodi, Paride Pelucchi.

**Formal analysis:** Alice Chiodi, Ettore Mosca.

**Funding acquisition:** Ettore Mosca.

**Methodology:** Alice Chiodi.

**Software:** Alice Chiodi, Ettore Mosca.

**Supervision:** Paride Pelucchi.

**Validation:** Paride Pelucchi.

**Visualization:** Ettore Mosca.

**Writing – original draft:** Alice Chiodi, Ettore Mosca.

**Writing – review & editing:** Alice Chiodi, Paride Pelucchi, Ettore Mosca.

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
