## [Decision Letter · Decision Letter 0]

15 Aug 2025

Dear Dr. Mosca,

Thank you for submitting your manuscript to PLOS ONE. After careful consideration, we feel that it has merit but does not fully meet PLOS ONE’s publication criteria as it currently stands. Therefore, we invite you to submit a revised version of the manuscript that addresses the points raised during the review process.

We look forward to receiving your revised manuscript.

Kind regards,

Mohammad H. Ghazimoradi

Academic Editor

PLOS ONE

2. We note that there is identifying data in the Supporting Information file < SupplementaryTables_3-11.xlsx>. Due to the inclusion of these potentially identifying data, we have removed this file from your file inventory. Prior to sharing human research participant data, authors should consult with an ethics committee to ensure data are shared in accordance with participant consent and all applicable local laws.

-Location data

Please remove or anonymize all personal information, ensure that the data shared are in accordance with participant consent, and re-upload a fully anonymized data set. Please note that spreadsheet columns with personal information must be removed and not hidden as all hidden columns will appear in the published file.

Reviewers' comments:

Reviewer's Responses to Questions

**Comments to the Author**

1. Is the manuscript technically sound, and do the data support the conclusions?

Reviewer #1: Yes

Reviewer #2: Yes

2. Has the statistical analysis been performed appropriately and rigorously?

Reviewer #1: Yes

Reviewer #2: Yes

3. Have the authors made all data underlying the findings in their manuscript fully available?

Reviewer #1: Yes

Reviewer #2: Yes

4. Is the manuscript presented in an intelligible fashion and written in standard English?

Reviewer #1: Yes

Reviewer #2: Yes

Reviewer #1: In this report the authors demonstrate a new method to quantify and analyze interactions among genes. Their methods proposes analysis of both intra-cellular and inter-cellular interactions. Methods to identify gene interaction networks have been previously describe and the authors correctly identify several algorithms (eg. STRING). The method presented here would add to existing ones. In this reviewer's opinion, it is valuable to disseminate this method in PLOS one. I believe that parallel techniques enrich our ability to analyze mechanisms of cellular transformation. However, I think that several improvements/clarifications should be made before dissemination. Below I suggest some:

1. The introduction is somewhat superficial in terms of summarizing existing methods. The authors mention STRING as another algorithm to test gene-gene interactions but do not really address how their method is different. I think the introduction should devote some space in the introduction briefly explaining other methods (STRING, KEGG, etc), then ending with a paragraph clearly explaining how this method is different.

2. The authors use gene expression data to validate their method. However, databases such as STRING uses multiple forms of data to create clusters: genetic interaction, physical interaction, gene mutation co-concurrence, etc. It is not clear to this reviewer whether the method described here can only be applied to gene expression data. This should be made clear somewhere.

3. Related to gene expression data, the authors use single-cell RNA data for TNBC. I understand why the authors did this because the data are cleaner with less likelihood of background noise. However, as the authors are probably aware, most gene expression data is provided by TCGA and can be downloaded as raw data or Z-transformed correlated to control tissue. Is this method able to analyze TCGA data as well or other forms of expression data from large repositories such as COSMIC and cBioPortal? Or is the method restricted to single cell analysis? Addressing this is important because most investigators use these repositories.

4. Finally, there has to be some form of control or perhaps comparison to already existing methods. If the authors run their raw data through STRING, KEGG, etc, do they get the same results? Or do they get different results? A conclusion should be made based on the outcome: if the same results, then this is just another method that can be used in lieu of existing ones; if different results, then once can say that it complements existing methods.

Reviewer #2: The paper proposes a framework, Ulisse, for enrichment analysis and cell–cell communication. The approach is well-structured; however, several aspects—such as pathway crosstalk quantification, ligand–receptor-based communication, and network-based DEG prioritization—overlap with existing tools like CellChat, NicheNet, and prior pathway crosstalk analyses. A significant limitation is the lack of comparison with existing tools. The S1 Table comparison is high-level and does not provide a quantitative benchmark or performance evaluation that would clearly demonstrate the framework’s superiority.

**Do you want your identity to be public for this peer review?** For information about this choice, including consent withdrawal, please see our Privacy Policy

Reviewer #1: No

Reviewer #2: No

---

## [Author Response · Author response to Decision Letter 1]

9 Sep 2025

RESPONSE TO REVIEWERS

Dear Editor,

we thank you and the reviewers for your assessments. We considered the reviewers comments and revised the manuscript accordingly. We revised the manuscript following the journal requirements. In relation to the issue about “identifying data”: please note that the file SupplementaryTables_3-11.xlsx does not include identifying data but aggregated data, generated by us based on publicly available data (doi: 10.6084/m9.figshare.17058077.v1). Below you will find our point-by-point answers (in red) to reviewers’ comments (in italic). We trust that we have adequately addressed the reviewers’ feedback.

Kind Regards,

Ettore Mosca

Reviewer #1

In this report the authors demonstrate a new method to quantify and analyze interactions among genes. Their methods proposes analysis of both intra-cellular and inter-cellular interactions. Methods to identify gene interaction networks have been previously describe and the authors correctly identify several algorithms (eg. STRING). The method presented here would add to existing ones. In this reviewer's opinion, it is valuable to disseminate this method in PLOS one. I believe that parallel techniques enrich our ability to analyze mechanisms of cellular transformation. However, I think that several improvements/clarifications should be made before dissemination. Below I suggest some:

Thanks for your assessment. We carefully considered your suggestions and revised our manuscript accordingly.

1. The introduction is somewhat superficial in terms of summarizing existing methods. The authors mention STRING as another algorithm to test gene-gene interactions but do not really address how their method is different. I think the introduction should devote some space in the introduction briefly explaining other methods (STRING, KEGG, etc), then ending with a paragraph clearly explaining how this method is different.

We revised introduction and discussion to clarify the state of the art, how Ulisse fits in it and the novel opportunities it offers. Please note that we used the STRING database as source of gene-gene interactions (sett methods “Molecular interactions and gene sets”).

2. The authors use gene expression data to validate their method. However, databases such as STRING uses multiple forms of data to create clusters: genetic interaction, physical interaction, gene mutation co-concurrence, etc. It is not clear to this reviewer whether the method described here can only be applied to gene expression data. This should be made clear somewhere.

We described a proof-of-concept using single-cell RNA sequencing, because this type of data accommodates both the quantification of pathway cross-talk and inter-cellular communication. However, our framework handles any gene-level score that can be meaningfully used to study the alterations in the interconnectivity among gene sets. We revised introduction and discussion to clarify this point.

3. Related to gene expression data, the authors use single-cell RNA data for TNBC. I understand why the authors did this because the data are cleaner with less likelihood of background noise. However, as the authors are probably aware, most gene expression data is provided by TCGA and can be downloaded as raw data or Z-transformed correlated to control tissue. Is this method able to analyze TCGA data as well or other forms of expression data from large repositories such as COSMIC and cBioPortal? Or is the method restricted to single cell analysis? Addressing this is important because most investigators use these repositories.

We revised the introduction to clarify the data types on which Ullise can be used. We revised the discussion to mention the applicability of Ulisse using data from public resources like those mentioned by the reviewer.

4. Finally, there has to be some form of control or perhaps comparison to already existing methods. If the authors run their raw data through STRING, KEGG, etc, do they get the same results? Or do they get different results? A conclusion should be made based on the outcome: if the same results, then this is just another method that can be used in lieu of existing ones; if different results, then once can say that it complements existing methods.

We revised introduction and discussion to clarify the state of the art, how Ulisse fits in it, the novel opportunities it offers, and why we leave a trustworthy quantitative benchmark for future work.

Collectively, compared to existing methods, Ulisse has a different focus, considers a more extensive statistical assessment, supports various input types and provides different outcomes. Such differences would lead to results that are more or less different from what can be obtained using other tools on the same input data. Therefore, considering the differences between Ulisse and existing tools and the issues in performing benchmarks in the absence of ground truths [e.g., see our recent review in Briefings in Bioinformatics 10.1093/bib/bbaf280], we decided to leave a comparison for future work. Indeed, a trustworthy comparison among tools in the absence of a ground truth should consider multiple approaches, like literature agreement, experimental validation, indirect validation, synthetic dataset validation and robustness assessment truths.

In this manuscript, we focused on describing Ulisse, its relation to existing approaches (extended in the revised manuscript), and showing the relevance of the results achieved in the proof-of-concept considering the literature on breast cancer.

Reviewer #2:

The paper proposes a framework, Ulisse, for enrichment analysis and cell–cell communication. The approach is well-structured; however, several aspects—such as pathway crosstalk quantification, ligand–receptor-based communication, and network-based DEG prioritization—overlap with existing tools like CellChat, NicheNet, and prior pathway crosstalk analyses. A significant limitation is the lack of comparison with existing tools. The S1 Table comparison is high-level and does not provide a quantitative benchmark or performance evaluation that would clearly demonstrate the framework’s superiority.

Thanks for your assessment, which helped us to improve our manuscript. We removed S1 Table, and extended introduction and discussion to clarify the state of the art, how Ulisse fits in it, the novel opportunities it offers, and why we leave a trustworthy quantitative benchmark for future work.

Collectively, compared to existing methods, Ulisse has a different focus, considers a more extensive statistical assessment, supports various input types and provides different outcomes. Such differences would lead to results that are more or less different from what can be obtained using other tools on the same input data.

For instance, the tools that you mentioned (CellChat and NicheNet) do not provide a network of communications among cell clusters that is based on quantifications at cell cluster level. Further, NicheNet works only on a built-in molecular interaction database.

Therefore, considering the differences between Ulisse and existing tools and, moreover, the issues in performing benchmarks in the absence of ground truths [e.g., see our recent review in Briefings in Bioinformatics 10.1093/bib/bbaf280], we decided to leave a comparison for future work. Indeed, a trustworthy comparison among tools in the absence of a ground truth should consider multiple approaches, like literature agreement, experimental validation, indirect validation, synthetic dataset validation and robustness assessment.

In this manuscript, we focused on describing Ulisse, its relation to existing approaches (extended in the revised manuscript), and showing the relevance of the results achieved in the proof-of-concept considering the literature on breast cancer.

---

## [Decision Letter · Decision Letter 1]

6 Oct 2025

Analysis of intracellular and intercellular crosstalk from omics data

PONE-D-25-19469R1

Dear Dr. Mosca,

We’re pleased to inform you that your manuscript has been judged scientifically suitable for publication and will be formally accepted for publication once it meets all outstanding technical requirements.

Kind regards,

Mohammad H. Ghazimoradi

Academic Editor

PLOS ONE

Additional Editor Comments (optional):

Reviewers' comments:

Reviewer's Responses to Questions

**Comments to the Author**

Reviewer #1: All comments have been addressed

2. Is the manuscript technically sound, and do the data support the conclusions?

Reviewer #1: Yes

3. Has the statistical analysis been performed appropriately and rigorously?

Reviewer #1: Yes

4. Have the authors made all data underlying the findings in their manuscript fully available?

Reviewer #1: Yes

5. Is the manuscript presented in an intelligible fashion and written in standard English?

Reviewer #1: Yes

Reviewer #1: The authors have been responsive to my critiques. This reviewer is satisfied. The revised version provides a more ind depth analysis.

**Do you want your identity to be public for this peer review?** For information about this choice, including consent withdrawal, please see our Privacy Policy

Reviewer #1: No

---

## [Editor Report · Acceptance letter]

PONE-D-25-19469R1

PLOS ONE

Dear Dr. Mosca,

I'm pleased to inform you that your manuscript has been deemed suitable for publication in PLOS ONE. Congratulations! Your manuscript is now being handed over to our production team.

Kind regards,

on behalf of

Dr. Mohammad H. Ghazimoradi

Academic Editor

PLOS ONE